# Hypofractionated Radiotherapy for Soft Tissue Sarcomas

**DOI:** 10.3390/cancers17071170

**Published:** 2025-03-31

**Authors:** Mehmet Murat Zerey, Amna Gul, Noah S. Kalman, Matthew D. Hall

**Affiliations:** Department of Radiation Oncology, Miami Cancer Institute, Herbert Wertheim College of Medicine, Florida International University, Miami, FL 33176, USA; mehmet.zerey@baptisthealth.net (M.M.Z.); amna.gul@baptisthealth.net (A.G.); noahk@baptisthealth.net (N.S.K.)

**Keywords:** radiation therapy, hypofractionation, stereotactic body radiation therapy, stereotactic ablative radiation therapy, proton therapy, SBRT, SABR, IMPT, STS, toxicities, resectable

## Abstract

Radiotherapy for soft tissue sarcomas (STS) is typically administered before or after surgery at daily doses of 1.8–2.0 Gy per fraction over 5–6 weeks. Hypofractionated radiotherapy, employing higher doses per fraction over a shorter overall course of treatment, has been used for metastatic disease and is increasingly being studied in the treatment of the primary tumor as an adjunct to surgery. This review examines the current evidence on the use of hypofractionated radiotherapy in patients with localized STS.

## 1. Introduction

Radiotherapy for soft tissue sarcomas (STS) is historically given using conventional fractionation of 1.8–2.0 Gy per fraction over 5–6 weeks. Aside from rhabdomyosarcoma and Ewing sarcoma in pediatric and young adult patients, most STS subtypes are radioresistant [1] and oncologic surgery with negative margins comprises the mainstay of curative therapy [2]. Due to the inherent radioresistance of most STS, hypofractionated radiotherapy may be beneficial to either shorten the duration of radiotherapy treatments or improve local control using dose-escalation strategies.

Hypofractionated radiotherapy employs higher radiation doses per fraction (typically ≥2.5 Gy per fraction) over a shorter course of treatment. This approach has emerged as an increasingly common treatment paradigm for breast and prostate cancers [3,4,5,6,7]. Moderate hypofractionation generally employs doses in the range of 2.5–3.0 Gy per fraction, leading to courses of radiation lasting several weeks. Ultrahypofractionation often uses even higher doses per fraction and can further shorten therapy to as few as 1–5 treatments. Daily doses of 6 Gy or higher are typically employed in these ultra-short courses of radiotherapy and have become increasingly popular for both palliative and curative intent radiotherapy in patients with small tumors in the brain and body.

In recent years, hypofractionated radiotherapy has been evaluated in the treatment of STS, a heterogeneous group of malignancies that pose significant challenges in terms of local control and long-term survival. Hypofractionated radiotherapy may offer similar or even superior outcomes compared to conventional radiotherapy regimens. In addition to reducing the overall treatment time and avoiding delays in surgical intervention or systemic treatment, hypofractionation may improve patient compliance and reduce healthcare costs, making it an attractive option for both patients and healthcare systems. This review aims to examine the current evidence on the use of hypofractionated radiotherapy in the treatment of localized STS, including its safety and efficacy. By synthesizing these findings, we seek to provide a comprehensive overview of the role that hypofractionated radiotherapy could play in future STS management.

## 2. Preoperative Hypofractionated Radiotherapy

In patients with either high-grade or recurrent STS, preoperative radiotherapy followed by surgical resection is the preferred treatment strategy whenever widely negative margins cannot be achieved without undue morbidity. Preoperative radiotherapy reduces the risk of late grade 2+ toxicities compared to postoperative radiation with a slight increase in the rate of wound healing complications, particularly in lower extremity tumors [8]. Preoperative hypofractionated radiotherapy over 1–3 weeks is increasingly being studied. One advantage is that a shorter course of treatment leads to the more rapid completion of preoperative radiotherapy, which may enable earlier surgical intervention. Several recent studies have investigated the effectiveness of preoperative hypofractionated radiotherapy for STS. These studies provide valuable insights into the feasibility and efficacy of this approach, highlighting both its promise and the challenges that remain in optimizing its use for STS patients. Below, we review and synthesize these published reports focusing on local control, pathologic response rates, and toxicities.

### 2.1. Non-Dose Escalation Studies

Kosela-Paterczyk et al. (2014) reported outcomes following preoperative hypofractionated radiotherapy in 272 patients with localized STS of the extremities and trunk. Patients received 25 Gy delivered in five fractions over consecutive days, followed by surgery within 2–4 days. The 3-year local recurrence-free survival and overall survival rates were 81% and 72%, respectively. A tumor size of >10 cm and high-grade tumors (G3) were associated with worse survival outcomes. The study reported acceptable early and late toxicity profiles, with 42% of patients experiencing some form of treatment-related toxicity; notably, the rates of toxicity were greater for lower-extremity tumors [9]. In 2021, updated results were provided on 311 patients. The primary endpoint was local recurrence-free survival. The authors reported a 5-year local recurrence-free survival rate of 81% and a 5-year overall survival rate of 63%. A total of 30.8% of patients experienced treatment-related adverse events, with 7.3% requiring surgery for wound complications. The authors concluded that preoperative hypofractionated radiotherapy was associated with local control rates comparable to conventional fractionation with manageable toxicities [10].

Kalbasi et al. published a phase II trial of neoadjuvant 30 Gy in five fractions over consecutive days in patients with high-grade STS using standard margins. The primary endpoint was the occurrence of grade 2 or higher late radiation toxicities; the incidence of major wound complications, local recurrences, and distant metastases were also reported. Among the 52 enrolled patients with a median follow-up of 29 months, 16% developed one or more grade 2+ late toxicities. In addition, 32% developed major wound complications. The study also identified several specific genetic biomarkers that were associated with higher rates of complications. The authors concluded that this regimen of 30 Gy in five fractions yielded a similar wound complication rate to conventional fractionation with reasonable late toxicities. The authors also found that this shorter regimen increased the utilization of neoadjuvant radiotherapy at a high-volume sarcoma center, suggesting that it may enable more patients to access treatment at a high-volume center if a shorter course of radiotherapy were available [11].

Parsai et al. reported the outcomes for sixteen patients treated with a median radiation dose of 30 Gy in five fractions. Surgery was performed within 7 days of radiotherapy. Ten patients (62.5%) achieved R0 resections (complete tumor removal with clear margins), while 6 patients had R1 resections. The median time interval from biopsy to completion of radiotherapy was 20 days, significantly shortening the overall treatment period compared to conventional regimens. No local failures were observed, with a median follow-up of 11 months. Wound healing complications occurred in 31% of patients and 19% required reoperation [12].

Potkrajcic et al. retrospectively analyzed the feasibility and safety of hypofractionated preoperative radiotherapy in the geriatric population. This study included 18 patients over 75 years of age who were unfit for standard fractionation radiotherapy due to frailty or medical comorbidities. Patients received 25 Gy in five fractions. Seventeen out of eighteen patients completed radiotherapy and surgery. Two patients developed simultaneous local and distant recurrences, while another two developed isolated distant recurrences, for a crude local control rate of 89%. The authors reported that hypofractionated radiotherapy may be a reasonable alternative in older patients, with comparable wound healing complications and similar oncologic outcomes to conventional preoperative radiotherapy [13].

Guadagnolo et al. reported outcomes for the HYPORT-STS trial, which employed a moderately hypofractionated 3-week course of preoperative radiotherapy. This single-arm, phase II trial included 120 patients who received 42.75 Gy in 15 fractions of 2.85 Gy/day over three weeks. The primary endpoint was the major wound complication rate within 120 days of surgery. The authors reported that 31% of patients developed major wound complications, which was similar to conventional fractionation. No grade 3 or higher acute radiation toxicities were reported; late radiation toxicities were limited to only 3% of patients. This 3-week hypofractionated regimen appeared safe and well tolerated with a low rate of severe late toxicities [14].

Table 1 summarizes the published data on the use of hypofractionated radiotherapy in STS with comparable dosing to conventional fractionation schedules.

### 2.2. Dose Escalation Studies

Bedi et al. published a phase 2 study of preoperative hypofractionated radiotherapy in patients with localized STS. The study involved 32 patients who were treated to 35 Gy in five fractions delivered every other day, followed by surgical resection 4–6 weeks after radiation. With a median follow-up of 36.4 months, the 3-year local control rate was 100%. The 3-year overall survival and distant metastasis-free survival rates were 82.2% and 69%, respectively. In total, 25% of patients developed major wound complications after surgery, while grade 2+ and grade 3 fibrosis was observed in 21.7% and 13% of patients, respectively [15].

Montero et al. reported the outcomes for 18 patients with STS of the limbs who received preoperative radiotherapy with a median dose of 52.5 Gy in 15 fractions; the range of daily dose per fraction was 3.3–4 Gy. The authors reported that 61.1% of patients achieved a favorable pathologic response, defined as ≥90% tumor necrosis; the pathologic complete response rate (pCR) was 36.8%. Forty-seven percent of patients experienced grade 1–2 acute skin toxicity and 38.8% developed wound complications. With a median follow-up of 14 months, no local relapses occurred. The 3-year overall survival and distant metastasis-free survival rates were 87% and 76.4%, respectively. The authors found that a favorable pathologic response was significantly associated with both improved 3-year overall survival and distant metastasis-free survival [16].

In a single-institution, prospective phase II trial, Leite et al. examined a stereotactic ablative radiotherapy regimen of 40 Gy in five fractions on alternate days, followed by surgery at least 4 weeks later. Of 25 patients enrolled, 20.8% had a pCR, and 28% developed wound complications. With a median follow-up of 20.7 months, the 2-year rates of local recurrence, distant metastasis, and cause-specific survival were 0%, 44.7%, and 10.6%, respectively. Notably, the authors reported a higher-than-expected amputation rate in this study, given that all patients were initially considered candidates for limb salvage surgery [17].

Kubicek et al. published another phase 2 study of stereotactic ablative radiotherapy in which 16 patients with extremity STS received 35–40 Gy in five fractions every other day. Median follow-up was 4.7 years. Acute grade 3 or higher toxicities were observed in one patient. Three patients (19%) developed wound complications after surgery. One patient developed a local recurrence, while five had distant recurrences [18].

Table 2 displays studies reporting on dose-escalated hypofractionated radiotherapy in STS.

## 3. Combined Neoadjuvant Approaches: Hypofractionated Radiotherapy and Chemotherapy or Hyperthermia

Recent studies on the use of neoadjuvant hypofractionated radiotherapy in combination with chemotherapy or hyperthermia to treat STS have shown promising results in terms of safety and efficacy. Combinatorial therapy is appealing for high-risk patients as it may lead to superior tumor downstaging and may reduce the risk of distant failure. Given the potential for greater toxicity, careful patient selection will be important to best evaluate whether this approach is valuable. Below, we reviewed the published data for combining hypofractionated radiotherapy with chemotherapy or hyperthermia for treatment intensification.

Ryan et al. reported outcomes in a single-arm, multicenter phase II trial involving 25 patients with high-risk STS of the extremity or trunk who received epirubicin and ifosfamide plus 28 Gy in 8 fractions during the second cycle of chemotherapy. Forty percent of patients achieved ≥95% pathologic necrosis at surgery. Significant acute toxicities developed, however, with 84% of patients experiencing at least one grade 4 toxicity, which were predominantly hematologic. In addition, 20% developed postoperative wound complications requiring surgery. The 2-year overall survival and disease-free survival rates were 84% and 62%, respectively. The authors concluded that, while this combined chemoradiotherapy regimen led to a promising pathologic response rate, the substantial toxicities (84% of patients developed one or more grade 4 acute toxicity) limited the patient tolerance of this aggressive regimen [19].

Pennington et al. reported outcomes in patients with extremity STS treated with neoadjuvant ifosfamide-based chemotherapy and hypofractionated preoperative radiotherapy, followed by limb-sparing surgery. A total of 116 patients received 28 Gy in eight fractions before surgery. The local recurrence rate was 11% at 3 years and 17% at 6 years; distant metastases developed in 25% of patients at 3 years and 35% at 6 years. Overall survival was 82% at 3 years and 67% at 6 years. On multivariate analysis, age > 60 years and tumor size >10 cm were significantly associated with worse overall survival [20].

Silva et al. investigated neoadjuvant hypofractionated radiotherapy (25 Gy in five fractions) combined with three cycles of doxorubicin and ifosfamide before surgery in 18 patients. At a median follow-up of 29 months, thirteen patients were alive without evidence of disease, one patient died due to metastatic disease, and three patients were alive with distant metastases. One patient developed a local relapse within the radiation field. Thirty-three percent of patients had a pCR, and thirty-three percent developed major wound complications [21].

Spałek et al. also combined preoperative hypofractionated radiotherapy (25 Gy in five fractions over 5 consecutive days) with doxorubicin-ifosfamide chemotherapy in patients with marginally resectable STS. In total, 46 patients received preoperative radiotherapy interdigitated with three cycles of chemotherapy. In the study, 72% of patients underwent R0 resections. Grade 3 or higher toxicities leading to dose reductions or treatment interruption occurred in fifteen patients (32.6%); wound complications were observed in eighteen patients (39.1%), with six having severe wound healing problems [22].

Qu et al. reported outcomes for 31 patients treated in 2021–2023 with 2–3 cycles of chemotherapy followed by hypofractionated radiotherapy (25–35 Gy at 5 Gy per fraction over 5–7 days) prior to surgery. Twenty-nine percent of patients experienced grade 3 or higher toxicities, and 25.8% developed postoperative wound complications. Most patients (96.7%) underwent limb preservation surgery. At a median follow-up of 20 months, the 1-year disease-free and overall survival rates were 79.3% and 89.6%, respectively [23].

In a separate cohort, Spałek et al. studied hypofractionated radiotherapy combined with regional hyperthermia in a phase II trial of patients with localized but marginally resectable or unresectable STS. Thirty patients enrolled, some of whom were not candidates for chemotherapy, either due to chemoresistant pathology, because they had progressed after neoadjuvant chemotherapy, or because of medical contraindications. Treatment consisted of 32.5 Gy in 10 fractions with four sessions of regional hyperthermia. In patients with unresectable disease and those who refused amputation, an additional 16 Gy in four fractions combined with two more hyperthermia sessions was delivered. The early local control rate was 97%, with only one patient developing rapid local recurrence post-surgery. Two patients developed distant metastases during follow-up. No severe hyperthermia-related adverse events were reported. The authors concluded that combined hyperthermia and hypofractionated radiotherapy was feasible in patients with marginally resectable STS [24].

Table 3 summarizes the studies reporting on hypofractionated radiotherapy in STS with concurrent chemotherapy or hyperthermia.

## 4. Adjuvant Hypofractionated Radiotherapy

The landmark O’Sullivan trial comparing preoperative and postoperative radiotherapy for STS demonstrated that higher rates of grade 2+ fibrosis and joint dysfunction occurred following postoperative radiotherapy. Given the higher doses and larger field sizes used in postoperative radiotherapy, published data for hypofractionation is limited [25]. Hypofractionated therapy remains of interest in selected patients who may not tolerate or accept longer courses of conventional radiotherapy, particularly in the elderly and medically frail. This approach may be attractive for patients who face difficulties with compliance, have significant comorbidities, or live a long distance from radiotherapy centers.

Soyfer et al. published a retrospective analysis of 21 elderly or medically unfit patients treated with hypofractionated adjuvant radiotherapy. Patients received between 39–48 Gy at 3 Gy per fraction over 13–16 sessions depending on margin status. At a mean follow-up of 26 months, local recurrences had developed in 14% of patients; all recurrences occurred in patients with close surgical margins (<3 mm). Eight patients (38%) developed distant metastases with three dying from metastatic disease during the study period. Only three patients experienced grade 2 or 3 dermatitis; the authors also reported low rates of delayed toxicities such as chronic pain and skin atrophy [26].

Hypofractionation may be reasonable in selected patients after limb conservation surgery. Further study will be crucial to ensure that acute and late toxicities remain acceptable in its implementation.

## 5. Hypofractionated Proton Therapy

Proton therapy significantly reduces the low and intermediate dose exposure of normal tissues surrounding the high-risk target volume, which can significantly reduce toxicities in selected patients [27]. Proton therapy has demonstrated promising toxicity reductions in pediatric patients receiving definitive intent radiotherapy to selected body sites, including the head and neck, trunk, abdomen, and pelvis [28], but its implementation for extremity tumors has been more limited. Here we summarize the current data and ongoing studies on proton therapy in adults treated for STS.

The PRONTO study is a prospective phase II trial evaluating the safety and efficacy of preoperative hypofractionated proton beam therapy (PBT) in patients with resectable extremity and truncal STS. The study will aim to treat 40 patients with 30 Gy radiobiological equivalent (RBE) in five fractions. This is the first study to investigate hypofractionated PBT for extremity STS, with the primary endpoint being the rate of major wound complications within 90 days of surgery. Secondary outcomes include late toxicities, local recurrence-free survival, distant metastasis-free survival, functional outcomes, and quality of life metrics. Data from this study may help to better understand the clinical benefits of reducing radiation exposure to bones, joints, and soft tissues for extremity tumors compared to modern highly conformal photon-based radiotherapy (NCT05917301) [29].

DeLaney et al. published a phase I trial of preoperative intensity-modulated proton therapy (IMPT) with a simultaneous integrated boost (SIB) for retroperitoneal sarcomas (RPS). This single-arm study enrolled 11 patients who received IMPT at a 50.4 GyRBE to the primary tumor volume and an escalated dose of up to 63 GyRBE in 28 fractions to the high-risk margin. Patients tolerated dose escalation up to the maximum planned dose of 63 GyRBE without dose-limiting toxicities. With a median follow-up of 18 months, no local recurrences were reported. Toxicities were limited, and there were no interruptions in radiation [30].

Based on these promising results, DeLaney et al. completed a phase II trial in RPS using this IMPT SIB regimen. Sixty patients received IMPT to 50.4 GyRBE in 28 fractions, with an escalated dose of 63 GyRBE to the high-risk region where close/positive margins were most likely. All patients completed preoperative IMPT. With a 23-month follow-up, two local recurrences were observed after surgery. Perioperative morbidity was consistent with published data for RPS. One case of bowel toxicity and neuropathy led to an amendment permitting a dose reduction of the SIB for these organs. Preliminary findings from this single-institution trial suggest that IMPT with dose escalation is feasible and may significantly improve local control compared to photon-based radiotherapy in RPS [31,32].

Yarlagadda et al. published a retrospective study of 16 patients treated in a single institution using DeLaney’s preoperative IMPT regimen followed by surgery. With a median follow-up of 18 months, the 3-year freedom from local failure rate was 68.2%, and the 3-year overall survival rate was 68.8%. No grade 3 or higher toxicities were observed. Mature results of the prospective trial and additional data will be important to evaluate whether this novel paradigm is useful and clinically reproduceable in RPS [33].

## 6. Conclusions

Hypofractionated radiotherapy has emerged as a versatile and effective option in the treatment of STS and may be beneficial to reduce treatment times and improve compliance in both the preoperative and adjuvant settings. The growing body of evidence suggests that preoperative hypofractionated radiotherapy can achieve favorable local control and survival outcomes while reducing the overall treatment burden in STS patients. When combined with chemotherapy or hyperthermia, hypofractionated radiotherapy demonstrates enhanced efficacy, and may benefit selected patients with high-risk features or marginally resectable primary tumors. The current data for postoperative hypofractionated radiotherapy remains limited in STS.

In this review, we sought to provide an overview of the current evidence regarding the outcomes and toxicities for this disease site. At present, while the volume of literature continues to expand, the length of follow-up in the majority of published studies remains short and the incidence of late effects and toxicities remains poorly characterized. As a result, caution should be taken with the implementation of hypofractionation into STS clinical practice, particularly in the most sensitive patients, such as the medically frail, where complications can be more severe and hazardous. Toxicities, including wound-healing complications, may be augmented with hypofractionated radiotherapy, particularly in the setting of dose escalation, ultrahypofractionated regimens employing ≤5–10 fractions, and when combined with systemic therapy. Ongoing reporting of the late toxicities and outcomes for STS patients treated with hypofractionated radiotherapy will be critical to best characterize the optimal patient demographics and tumor characteristics for this approach and the expected outcomes. The introduction of proton therapy adds another promising technology, which can minimize radiation exposure to healthy tissues and may potentially reduce long-term complications. As research continues to refine the use of hypofractionated radiotherapy in STS, this approach has the potential to improve both oncologic outcomes and patient quality of life, positioning it as a key modality in the evolving treatment landscape for STS.

## Figures and Tables

**Table 1 cancers-17-01170-t001:** Non-Dose Escalation Studies of Hypofractionated Radiotherapy.

**Study**	**Year**	** *n* **	**Dose/Fractionation**	**BED (Gy)**	Radiation Technique	Time to Surgery After RT	% of Patients Receiving Chemotherapy	R0 Resection Rate (%)	Local Control	Toxicity	Wound Complications
Kosela-Paterczyk [9].	2014	272	25 Gy/5 fx (daily)	56	3DCRT	3–7 days	22	79	81% 3-year	15% late G2	32% (7% requiring reoperation)
Kalbasi [11]	2020	52	30 Gy/5 fx (daily)	75	IMRT, 3DCRT, electron	2–6 weeks (median 4)	12	82	94% 2-year	16% late G2	32% requiring reoperation
Parsai [12]	2020	16	30 Gy/5 fx (daily)	75	IMRT, VMAT	0–7 days (median 1)	0	62	100% 10-month	0% ≥ G3 late toxicity	31% (19% requiring reoperation)
Kosela-Paterczyk [10]	2021	311	25 Gy/5 fx (daily)	56	3D-CRT, IMRT, VMAT	2–4 days	30	84	81% 5-year	8.6% late toxicity	24%
Potkrajcic [13].	2021	18	25 Gy/5 fx (daily)	56	3DCRT, IMRT, VMAT	15–45 days (median 29)	0	72	92% 6-month	0% > G1 toxicity	29% requiring reoperation
Guadagno [14].	2022	120	42.75 Gy/15 fx (daily)	67–83	3DCRT, IMRT, VMAT, electron, proton	4.6–6.4 weeks (median 5.7)	33	90	93% 30-month	0% > G3 late toxicity	31% (10% requiring reoperation)

**Table 2 cancers-17-01170-t002:** Dose Escalation Studies of Hypofractionated Radiotherapy.

**Study**	Year	*n*	Dose/Fractionation	BED (Gy)	Radiation Technique	Time to Surgery After RT	% of Patients Receiving Chemotherapy	R0 Resection Rate (%)	Local Control	Toxicity	Wound Complications
Bedi [15]	2022	32	35 Gy/5 fx (every other day)	96	3DCRT, IMRT	19–67 days (median 41)	31	91	100% 3-year	22% G2 fibrosis, 13% G3 fibrosis	25%
Montero [16]	2023	18	52.5 Gy/15 fx (daily)	98	3D-CRT, IMRT, VMAT	3–21 weeks (median 7)	33	94	100% 3-year	44% G1 or G2 acute skin toxicity	38% requiring reoperation
Leite [17]	2021	25	40 Gy/5 fx (every other day)	120	SBRT, IMRT, VMAT	27–91 days (median 60)	20	96	100% 2-year	6% late G2 fibrosis, 6% late G2 stiffness, 11% late G2 edema	28%
Kubicek [18]	2021	16	35 or 40 Gy/5 fx (every other day)	96–120	SBRT	29–83 days (median 41)	19	75	93% 4.7-year	27% late G1–2, 7% late G4	20%

**Table 3 cancers-17-01170-t003:** Hypofractionated Radiotherapy Combined with Chemotherapy or Regional Hyperthermia.

**Study**	**Year**	** *n* **	**Dose/Fraction and (Frequency)**	**BED (Gy)**	**Radiation Technique**	Time to Surgery After Treatment	Chemotherapy	R0 Resection Rate (%)	Local Control	Toxicity	Wound Complications
Ryan [19]	2008	25	28 Gy/8 fx (daily)	N/A	N/A	N/A 5–19 weeks (median 10) after chemotherapy initiation	Epirubicin and ifosfamide	88%	88% 2-year	84% G4 acute toxicity	20% requiring reoperation
Pennington [20]	2018	116	28 Gy/8 fx (daily)	52	3D-CRT	1–2 weeks	Doxorubicin and ifosfamide	93%	83% 6-year	15% acute and late toxicity	11%
Silva [21]	2021	18	25 Gy/5 fx (daily)	56	3D-CRT, IMRT	2.4–15.7 weeks (median 6)	Doxorubicin and ifosfamide	83%	95% 29-month	6% late G2 fibrosis, 6% late G2 stiffness, 11% late G2 edema	33%
Spałek [22]	2021	46	25 Gy/5 fx (daily)	56	3D-CRT, VMAT, IMRT	22–28 days (median 25)	Doxorubicin and ifosfamide	72%	67% 2-year	9% G2 acute skin toxicity	34%
Qu [23]	2024	31	25–35 Gy/5 fx (daily)	N/A	IMRT	1–2 weeks	N/A	93%	93% 22-month	29% G3 acute toxicity	26%

## Data Availability

No new data were created or analyzed in this review, which is a synthesis of published data. Data sharing is not applicable to this article.

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
