# Peer review of "Hypofractionated Radiotherapy for Soft Tissue Sarcomas"

_cancers, 2025, doi:10.3390/cancers17071170_

Round 1

Reviewer 1 Report

Comments and Suggestions for Authors

The revision of the article  Hypofractionated Radiotherapy for Soft Tissue Sarcomas

led to these final considerations:

  • The Introduction is clear and complete
  • The revision of the literature is accurate and well presented in different sub chapters.

In my opinion the aspects that should be improved  and discussed   at best are

1) There is a great dissimilarity among the reported studies : different protocols, different administered  doses ( 25 to 30 Gy), days of treatment ranging from 5 to 15. Hypofractionation is a term but the meaning is very different in the different  reported studies

2) Grade 3 -4 toxicities ranges from 30% to 80% if combined with Chemotherapy.   Correctly  the Authors  say  that acute toxicity is high. Difficult to affirm  that is manageable.

3) Patients selection should be accurately performed  and frail and eldely Patients cannot be considered , differently from what was said by  Potkrajcic

4) In my opinion all these points should be accurately stressed in  a  specific  Discussion paragraph, or the Conclusions need to be extended from the current form.

Author Response

The revision of the article Hypofractionated Radiotherapy for Soft Tissue Sarcomas

led to these final considerations:

  • The Introduction is clear and complete
  • The revision of the literature is accurate and well presented in different sub chapters.

In my opinion the aspects that should be improved and discussed   at best are

  • There is a great dissimilarity among the reported studies: different protocols, different administered  doses ( 25 to 30 Gy), days of treatment ranging from 5 to 15. Hypofractionation is a term but the meaning is very different in the different  reported studies

Thank you for the comment.  We agree that the regimen used varies across published studies.  Radiotherapy delivered using 5 fractions or fewer are generally called SBRT when using doses considered to be ablative in nature (generally defined as >=6 Gy per fraction), although there is some controversy on this point, as the extra-nuclear cell kill mediated by acid sphingomylinase seems to arise/accelerate in vitro above 8 Gy per fraction.  Such regimens may be called ulta-hypofractionation.  Hypofractionation, in general, is delivery of radiation using fractions that are larger than 2 Gy per fraction (conventional fractionation), with intent to reduce the total time course of radiotherapy. 

We have added this definition/clarification in the Introduction, Paragraph 2.  We have also added a discussion of this nuance in the Conclusions. 

  • Grade 3 -4 toxicities ranges from 30% to 80% if combined with Chemotherapy.   Correctly  the Authors  say  that acute toxicity is high. Difficult to affirm  that is manageable.

Thank you.  This is a very good point and an oversight on our part.  We have edited to state that caution need be applied in combining chemoradiotherapy in this setting due to resulting toxicities. 

  • Patients selection should be accurately performed  and frail and eldely Patients cannot be considered , differently from what was said by  Potkrajcic

Thank you for the suggestion.  We agree that limited data exists specifically looking at elderly patients treated for STS, but older patients comprise a large number of sarcoma patients. 

To address this concern, we added a sentence to the paragraph on the study by Potkrajcic and colleagues indicating that caution is needed in treating these patients who are at higher risk of potential toxicity.  We added a similar caveat to the Conclusions.  While this population is definitely one where a shorter course of treatment would be ideal, care must be taken to mitigate harm as data emerges. 

  • In my opinion all these points should be accurately stressed in  a  specific  Discussion paragraph, or the Conclusions need to be extended from the current form.

Thank you, we have specifically addressed all of the above in the Conclusions, which are now expanded. 

Reviewer 2 Report

Comments and Suggestions for Authors

In the introduction stating that all but 2 sarcomas is radioresistant is somewhat misleading. While most sarcomas do not have radiographic response this is not the goal for the use of radiation in sarcoma. It is an important part of limb salvage strategies.

 In line 50-51 should read: “In patients with high grade STS, preoperative radiotherapy followed by surgical resection is the preferred treatment strategy.” Radiation is standard of care for high grade sarcomas. It is misleading to state that it is used only when negative margins cannot be achieved. I would not lump high grade STS and recurrent here. These are two different clinical scenarios.

  1. Preoperative Hypofractionated Radiotherapy – the content of this paragraph is somewhat redundant to the prior. It should be revised to minimize these redundancies.

A summary of the data presented in sections 2 and 3 would be useful to discuss how it applies as is done in section 4.

Given the data presented in most of the studies the conclusion that hypofractionation can provide comparable local control to conventional cannot be drawn. The majority of studies focused on wound complications and follow-up was too short to assess local failure rate.

Agree with other reviewer that the tables should be referenced in the text.

Reviewer 3 Report

Comments and Suggestions for Authors

Nice review.

  1. Would add a section on the histologic response described in the papers included in Tables 1-3.  Can one suggest that tumor viability is similar or better than with conventional fractionation?
  2. Are there ongoing or planned trials (phase II or III) to compare hypo to conventional fractionation?  If not, how would you design one? Is one needed?
  3. Is there evidence that certain histologies respond better to hypofractionation?

Author Response

Nice review.

  1. Would add a section on the histologic response described in the papers included in Tables 1-3.  Can one suggest that tumor viability is similar or better than with conventional fractionation?

This is a great comment.  We went back and reviewed the individual manuscripts.  With regret, the majority of papers did not report the pathologic complete response or necrosis rate at surgery.  Given this limited reporting, we feel that adding them to the Tables will not be overly beneficial since most entries would be blank. 

In principle, the expectation is that non-dose escalated radiotherapy should have comparable pathologic response rates to standard fractionation.  This is because tumor cell kill from radiation is both a function of the total dose of radiation given, but also the dose per fraction.  Most hypofractionated radiotherapy regimens are crafted to lead to a comparable tumor cell kill as a standard fractionation regimen.  This is particularly true for breast cancer and sarcoma, where the response of tumor to radiation is similar to the underlying normal soft tissues.  As a result, the expectation for non-dose escalated regimens is that tumor cell kill and soft tissue toxicity are expected to be comparable.  We hypothesize that this may have led investigators to de-emphasize reporting of this endpoint in their manuscripts (unfortunately).

We think this is useful information to report however, so wherever possible, we have verified that pathologic response rates/necrosis rates are reported in the summary paragraphs when provided in the papers. 

  1. Are there ongoing or planned trials (phase II or III) to compare hypo to conventional fractionation?  If not, how would you design one? Is one needed?

To our knowledge, we are not familiar with any planned Phase II-III randomized trials comparing fractionation regimens in STS. 

Due to the limited number of patients treated even at high volume centers for STS, it would be difficult to complete this trial in a reasonable time interval.  While this is an easy concept that has been done several times in breast cancer, all of which have generally shown very comparable toxicity and efficacy when non-dose escalated strategies are employed, the incidence of breast cancer and the success of enrolling such patients on NSABP/NRG cooperative group trials and even at large institutions present a challenge in STS. 

One author from our group recently applied for a position on the new Soft Tissue Sarcoma Subcommittee of NRG Oncology and proposed a Phase II study within the cooperative group to expand knowledge in this space.  The proposal used a single arm design (randomization was not proposed as it would have doubled the number of patients) to examine two hypofractionated radiotherapy regimens lasting 5 and 15 fractions stratified by tumor size and location.  The primary aim was to measure acute toxicity and late toxicity at 3 years taking a non-inferiority approach to published data from conventional fractionation treatments of STS.  A secondary aim was to examine the outcomes across different institutions stratified by volume, with the hypothesis that larger institutions may have better outcomes than smaller ones, which may suggest that broad implementation of hypofractionation would have to be performed carefully.  This non-inferiority design has been used successfully in several breast cancer trials.  One example is here from Lancet Oncology:

Hypofractionated versus conventional fractionated postmastectomy radiotherapy for patients with high-risk breast cancer: a randomised, non-inferiority, open-label, phase 3 trial

Wang, Shu-Lian et al.

The Lancet Oncology, Volume 20, Issue 3, 352 – 360. March 2019

  1. Is there evidence that certain histologies respond better to hypofractionation?

Great question.  Due to the large number of sarcoma histologies, I do not think this question has been tackled for the majority of soft tissue sarcomas in a concerted manner.  There was a Phase 2 study from Mayo Clinic that demonstrated greater responsiveness of Ewing sarcoma to SABR (an ablative hypofractionation regimen) than osteosarcoma.  Ewing sarcoma and Rhabdomyosarcoma though are considered radiosensitive, whereas Osteosarcoma, like most adult STS, are not.  There is also data that chordomas, which are particularly radiation resistant, may respond better to hypofractionation vs. conventional fractionation.  However, for the vast majority of STS, the tumor radiosensitivity is believed to be similar to that of the normal surrounding tissues as modeled by the alpha/beta ratio, meaning that hypofractionation is not expected to alter the tumor response compared to a conventional fractionation schedule of comparable intensity. 

Thank you so much for reviewing our manuscript. 

Round 2

Reviewer 3 Report

Comments and Suggestions for Authors

thank you for your response.  I agree that hypofractionation is an important trend in radiation oncology, although I am struck by the relatively poor quality of the trials (more a compilation of institutional experience).  As a non-radiation oncologist, I would like to see more studies in the neoadjuvant setting that focus on comparisons between traditional fractionation and any one of the hypofractionated regimens in a particular histology to finally nail down the conclusion that conventional fractionation is no better.  This was finally done in breast cancer (although we have not completely eradicated the old 6-week regimen (possibly due to reimbursement issues)).